# PROBE: PROBING RESIDUAL CONCEPT CAPACITY IN ERASED TEXT-TO-VIDEO MODELS

## ABSTRACT

Text-to-video (T2V) diffusion models have achieved remarkable progress in generating temporally coherent, high-quality videos. However, their ability to generate sensitive or undesired concepts has raised concerns, motivating the development of concept erasure techniques that aim to suppress specific semantics while preserving general generation quality. Despite rapid progress in text-to-image (T2I) concept erasure, the effectiveness of these methods in T2V settings remains largely unquantified. In this work, we introduce PROBE, a systematic framework to measure the residual capacity of erased T2V models to represent and regenerate a target concept. PROBE learns a compact token embedding by jointly optimizing across all frames and timesteps, augmented with a latent alignment loss that anchors the recovered representation to the spatiotemporal structure of the original concept. The resulting embedding serves as a reusable probe that enables reproducible, large-scale robustness evaluation across different erasure methods and models. Experiments on multiple T2V architectures demonstrate that PROBE reveals substantial residual concept capacity even after erasure, providing new insights into the limitations of existing techniques and establishing a principled benchmark for future research on safe video generation.

## 1 INTRODUCTION

Text-to-image (T2I) generative models have achieved remarkable progress with large-scale diffusion architectures such as Stable Diffusion (Rombach et al., 2022) and DALL·E (Ramesh et al., 2021). This rapid progress has also made *concept erasure*—the selective suppression of unwanted or unsafe semantics in pretrained models—an increasingly important direction for ensuring model safety, copyright compliance, and responsible deployment. A diverse set of approaches has been developed to address this problem, including text encoder editing (Zhang et al., 2024b), cross-attention steering (Gandikota et al., 2024), and UNet fine-tuning (Chavhan et al., 2025). These methods have demonstrated that it is possible to selectively disable targeted concepts while largely preserving the model's generative capabilities.

Despite these advances, the problem of concept erasure in text-to-video (T2V) models remains largely underexplored, even as T2V systems such as CogVideoX (Yang et al., 2025), Hunyuan-Video (Kong et al., 2024), and Open-Sora (Zheng et al., 2024) achieve unprecedented levels of visual quality and temporal coherence. Compared to images, videos introduce unique challenges: temporal dependencies propagate residual traces across frames, motion acts as an additional semantic carrier, and longer sequence lengths amplify even subtle misalignments or leakage. These factors make it substantially more difficult to guarantee that a concept has been fully removed. Yet existing work on T2V erasure (Ye et al., 2025; Yoon et al., 2025; Liu & Tan, 2025) has so far focused primarily on suppressing visible occurrences of the concept in prompted outputs, leaving open the question of whether erasure is robust to distribution shifts, paraphrased prompts, or latent reactivation.

Evidence from the T2I literature suggests that such robustness cannot be taken for granted. A growing body of research has revealed that erased concepts can often be recovered through adversarial prompting, paraphrased descriptions, or more recently, textual inversion methods that learn pseudo-tokens capable of reintroducing the suppressed concept (Chin et al., 2024; Tsai et al., 2024; Pham et al., 2024). While these findings have raised important concerns in the image domain, whether similar vulnerabilities exist in T2V models is largely unknown. This is a critical gap: without standardized probes and benchmarks, developers may overestimate the reliability of erasure, potentially

Figure 1: Vulnerability of concept-erasure defenses on the *golf ball* concept. We compare three representative methods (*NegPrompt*, *SAFREE*, and *T2VUnlearning*). Top: model outputs after erasure, where the target concept is largely suppressed. Bottom: applying our PROBE attack reliably reactivates the erased concept.

leading to unsafe model releases. Moreover, the temporal dimension of video introduces new failure modes that are invisible in frame-level evaluation—for instance, concepts may reappear partway through a generated clip or exhibit temporal drift that gradually reconstructs the erased semantics.

To address this gap, we propose **PROBE** (**Prob**ing r**E**sidual concept capacity in erased text-to-video models), a principled framework for quantifying how much concept information remains encoded after erasure (i.e., the model's residual ability to reproduce the removed concept). Rather than treating erasure as a binary outcome, PROBE learns a compact pseudo-token that can be inserted into prompts to maximally activate the erased model's residual representation of a target concept. To explicitly address temporal drift and frame-wise inconsistency, we introduce a latent alignment loss that complements per-step reconstruction, enforcing clip-level semantic consistency and preventing degenerate solutions that ignore concept semantics. This combination enables continuous, reproducible, and semantically faithful quantification of residual capacity without modifying model parameters. We systematically evaluate PROBE on two state-of-the-art T2V models—CogVideoX-2B and CogVideoX-5B—and three representative erasure strategies: NegPrompt, an input-level conditioning method; SAFREE, an activation steering approach; and T2VUnlearning, a parameter-efficient unlearning technique. As illustrated in Fig. 1, PROBE reliably bypasses all three defenses: while the erased models suppress the target concept, our pseudo-token reactivates it with high fidelity. Across all combinations of models and erasure methods, PROBE reveals measurable residual concept capacity, showing that even the strongest current defenses do not completely remove the target concept but instead rely on partial suppression. These findings highlight the need for standardized probing tools to provide faithful, reproducible estimates of erasure robustness.

Our contributions are summarized as follows:

- **A systematic framework for T2V concept probing.** We present the first principled framework for quantifying residual concept capacity in T2V models, extending concept inversion from the image domain to the video domain.

- **Latent alignment for robust reactivation.** We propose a latent alignment loss that strengthens concept reactivation and mitigates temporal drift, leading to more reliable and semantically faithful assessment of erasure robustness.

- **Comprehensive evaluation across erasure paradigms.** We benchmark three representative T2V erasure strategies—NegPrompt, SAFREE, and T2VUnlearning—across multiple models and show that all leave measurable residual capacity, underscoring the need for standardized evaluation protocols.

## 2 RELATED WORK

### 2.1 FROM T2I TO T2V GENERATIVE MODELS

The success of text-to-image (T2I) models such as Imagen (Saharia et al., 2022) and Stable Diffusion (Rombach et al., 2022) has driven rapid advances in text-to-video (T2V) generation. Motivated by these breakthroughs, early T2V approaches extended U-Net backbones with cross-attention into the temporal domain (Ho et al., 2022; Wu et al., 2023; Wang et al., 2023), laying the groundwork

for controllable video synthesis with temporal coherence. More recent work has adopted Diffusion Transformers (DiT) (Peebles & Xie, 2023), which process text and video tokens in a unified attention space. CogVideoX (Yang et al., 2025) employs multimodal DiTs to improve efficiency and alignment, and HunyuanVideo (Kong et al., 2024) explores both dual-stream and single-stream tokenization strategies. While these advances position T2V as a powerful tool for storytelling and immersive media, they also underscore its inherited ability from T2I to generate sensitive content, motivating research on concept erasure.

## 2.2 Concept Erasure in T2I Models

To mitigate unsafe or sensitive generations, researchers have proposed concept erasure, which seeks to suppress unwanted semantics while preserving the general utility of the model (Zhang et al., 2024a). In the T2I domain, methods can be grouped by intervention level (Xie et al., 2025). The first category edits text encoders or embeddings (Cywiński & Deja, 2025; Sridhar & Vasconcelos, 2024). The second modifies cross-attention maps to weaken text–image alignment (Lee et al., 2025; Lu et al., 2024). The third tunes or prunes the U-Net backbone (Huang et al., 2024; Kim et al., 2024). The fourth introduces lightweight adapters or closed-form parameter edits (Lyu et al., 2024; Gandikota et al., 2024). Representative examples include ESD (Gandikota et al., 2023) and MACE (Lu et al., 2024). Although these methods have demonstrated effectiveness in suppressing targeted concepts while largely preserving model utility, extending them to T2V introduces additional challenges, as temporal coherence can propagate residual traces across frames, motion dynamics provide alternative cues, and the high dimensionality of video sequences increases the cost of fine-tuning.

## 2.3 Concept Erasure in T2V Models

Research on T2V concept erasure is still at an early stage, with only a few initial studies (Liu & Tan, 2025; Yoon et al., 2025; Ye et al., 2025; Xu et al., 2025). Preliminary efforts can be grouped into several directions, including training-free strategies that adjust input prompt embeddings (Yoon et al., 2025; Xu et al., 2025), parameter-efficient fine-tuning such as LoRA applied to attention layers (Ye et al., 2025), and loss-based optimization strategies that update the text encoder to weaken alignment with sensitive concepts (Liu & Tan, 2025). Despite these explorations, current methods remain exploratory and lack standardized benchmarks or systematic robustness evaluation, particularly under adversarial prompting or long-sequence generation. These limitations suggest that achieving reliable and comprehensive concept erasure in T2V models remains an open challenge.

## 3 Method

Our objective is to systematically measure whether a text-to-video (T2V) model that has undergone concept erasure still retains residual capacity to represent a target concept $c^*$. Rather than treating erasure as a binary event, we aim to quantify the degree of residual representation by learning a compact embedding $v^*$ that maximally elicits the model's internal response to $c^*$. We refer to our framework as PROBE (Probing Residual Concept Capacity). This section first reviews diffusion model preliminaries and textual inversion, then formalizes erased-concept recovery, and finally introduces our PROBE framework and training objective.

### 3.1 Preliminaries

**Text-to-Image Diffusion.** A T2I diffusion model defines a forward noising process that gradually perturbs a clean image $x_0$ into Gaussian noise through

$$q(x_t|x_{t-1}) = \mathcal{N}(x_t; \sqrt{\alpha_t}x_{t-1}, (1 - \alpha_t)I), \tag{1}$$

where $\alpha_t$ is a variance schedule. The reverse process is parameterized by a denoiser $\epsilon_\theta$ that predicts the added noise $\epsilon$ given $x_t$ and text embedding $c$:

$$p_\theta(x_{t-1}|x_t, c) = \mathcal{N}(x_{t-1}; \mu_\theta(x_t, c, t), \sigma_t^2 I). \tag{2}$$

Training minimizes the denoising loss:

$$\mathcal{L}_{\text{T2I}} = \mathbb{E}_{x_0,c,t,\epsilon}\big[\|\epsilon - \epsilon_\theta(x_t, c, t)\|_2^2\big]. \tag{3}$$

**Text-to-Video Diffusion.** T2V generalizes this to spatiotemporal latents $x_0 \in \mathbb{R}^{T \times H \times W \times C}$, where $T$ is the number of frames. The denoiser $\epsilon_\theta$ is replaced by a spatiotemporal U-Net or Diffusion

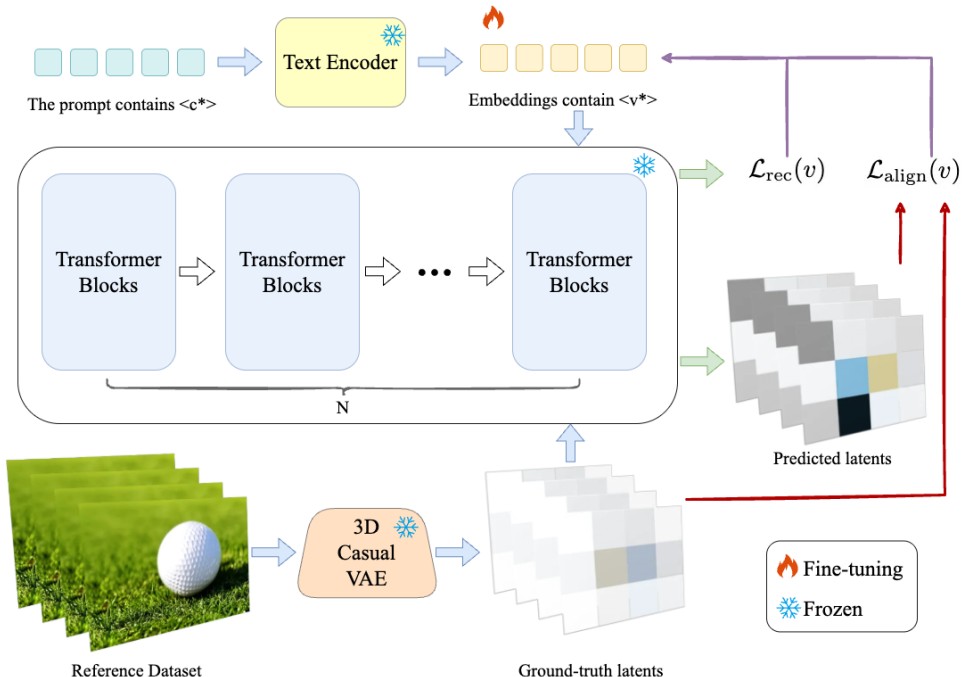

Figure 2: The architectural of our PROBE framework.

Transformer that models spatial and temporal dependencies jointly:

$$\mathcal{L}_{\text{T2V}} = \mathbb{E}_{x_0,c,t,\epsilon}\big[\|\epsilon - \epsilon_\theta(x_t, c, t)\|_2^2\big]. \tag{4}$$

This coupling enforces temporal coherence, which makes concept erasure and its evaluation more challenging than in the T2I setting.

**Concept Inversion in T2I.** Textual inversion learns a placeholder token $\tilde{c}$ with embedding $v$ that reconstructs a target concept $c^*$. Given latent $z_0 = \mathcal{E}(x_0)$ from encoder $\mathcal{E}$, and noisy latent $z_t = \alpha_t z_0 + \sigma_t \epsilon$, the embedding is optimized as

$$v^* = \arg\min_v \mathbb{E}_{z_t,\epsilon,t}\big[\|\epsilon - \epsilon_\theta(z_t, v, t)\|_2^2\big]. \tag{5}$$

This results in an embedding that reintroduces $c^*$ into the model's vocabulary without modifying $\theta$. In this work, we extend this idea to video, but with additional design choices to address temporal consistency and semantic faithfulness.

### 3.2 PROBLEM FORMULATION: QUANTIFYING RESIDUAL CONCEPT CAPACITY

Let $\epsilon_\theta$ be a T2V denoiser after erasure of $c^*$. We assume partial erasure, meaning residual information about $c^*$ may still be encoded in the model. Our goal is to find an embedding $v^*$ that maximally elicits this residual representation.

Given a set of clips $\{x_0^{(i)}\}$ containing $c^*$, we encode them into spatiotemporal latents $z_0^{(i)} = \mathcal{E}(x_0^{(i)})$ and construct noisy latents

$$z_t^{(i)} = \alpha_t z_0^{(i)} + \sigma_t \epsilon, \quad \epsilon \sim \mathcal{N}(0, I). \tag{6}$$

We then introduce a placeholder token $\tilde{c}$ with learnable embedding $v \in \mathbb{R}^d$ and solve

$$v^* = \arg\min_{v \in \mathbb{R}^d} \mathbb{E}_{i,t,\epsilon}\big[\|\epsilon - \epsilon_\theta(z_t^{(i)}, v, t)\|_2^2\big] + \lambda \mathcal{L}_{\text{align}}(v). \tag{7}$$

The learned $v^*$ provides a compact probe: conditioning generation on $\tilde{c}$ produces videos that reveal the model's residual capacity to represent $c^*$.

### 3.3 PROBE: A UNIFIED FRAMEWORK FOR QUANTIFYING RESIDUAL CONCEPT CAPACITY

Evaluating concept erasure in text-to-video (T2V) models is fundamentally more challenging than in the image domain. Naively checking whether a prompt still generates $c^*$ yields a binary "erased / not erased" judgment, which is inadequate for measuring partial suppression or subtle re-emergence of the concept under adversarial conditions. Moreover, simple per-frame concept inversion often produces temporally incoherent results and fails to exploit the full spatiotemporal information present in videos. Finally, there is no standardized mechanism to compare erasure methods under consistent conditions, making reproducibility and benchmarking difficult.

To address these gaps, we introduce PROBE, a unified framework that learns a single, compact embedding $v^*$ serving as a standardized probe of residual concept capacity. Unlike previous approaches that optimize frame-wise embeddings or rely solely on reconstruction loss, PROBE introduces joint spatiotemporal optimization, a latent alignment term that anchors $v^*$ to the semantic content of $c^*$, and a token distillation step that enables large-scale, reproducible benchmarking across models and erasure methods. Together, these components turn concept inversion from a heuristic attack into a principled, quantitative evaluation tool.

**Joint Spatiotemporal Optimization.** A key observation is that optimizing a separate embedding $v_t$ for each frame leads to flicker and instability: different frames may converge to embeddings that encode slightly different semantic variations, yielding temporally inconsistent generations. To prevent this, PROBE optimizes a single shared embedding $v$ jointly across all frames and timesteps:

$$\mathcal{L}_{\text{rec}}(v) = \mathbb{E}_{i,t,\epsilon}\big[\|\epsilon - \epsilon_\theta(z_t^{(i)}, v, t)\|_2^2\big], \tag{8}$$

where $z_t^{(i)}$ is the noised latent of clip $i$ at timestep $t$, and $\epsilon_\theta$ is the denoiser of the erased T2V model. By sharing gradients across timesteps and frames, this formulation encourages $v$ to capture a coherent clip-level representation of $c^*$, improving both temporal consistency and optimization stability.

**Latent Alignment for Semantic Grounding.** While $\mathcal{L}_{\text{rec}}$ enforces that $v$ follows the correct denoising trajectory, it does not guarantee that the resulting generations are semantically aligned with $c^*$. In practice, we observed that reconstruction-only optimization may produce embeddings that simply reduce noise variance, resulting in visually plausible but semantically irrelevant outputs. To address this, we introduce a latent alignment term:

$$\mathcal{L}_{\text{align}}(v) = \mathbb{E}_{i,t}\big\|\hat{z}_0^{(i)}(v,t) - z_0^{(i)}\big\|_2^2, \tag{9}$$

where $\hat{z}_0^{(i)}(v,t) = (z_t^{(i)} - \sigma_t \cdot \epsilon_\theta(z_t^{(i)}, v, t))/\alpha_t$ is the model's prediction of the clean latent under embedding $v$. This loss anchors the learned embedding to the true spatiotemporal structure of $c^*$, discouraging degenerate solutions that minimize noise error without reactivating the concept. In Section 4.4, we present ablation studies on the erased concept *nudity*. The results show that incorporating $\mathcal{L}_{\text{align}}$ substantially increases the recovered nudity rate, highlighting its critical role in achieving meaningful and faithful concept recovery.

**Unified Training Objective.** Combining the two components yields the final objective:

$$\mathcal{L}_{\text{total}}(v) = \mathcal{L}_{\text{rec}}(v) + \lambda\,\mathcal{L}_{\text{align}}(v), \tag{10}$$

where $\lambda > 0$ is a weighting factor that balances local denoising fidelity with global semantic alignment. A small $\lambda$ prioritizes noise reconstruction but risks under-representing $c^*$, while a large $\lambda$ enforces stronger semantic recovery at the cost of higher reconstruction error. We select $\lambda$ via a grid search on a held-out validation set, using concept similarity metrics as the selection criterion.

**Token Distillation for Reproducibility.** Once $v^*$ is learned, we distill it into a reusable pseudo-token `<c*>` that can be prepended to any text prompt. This step converts the recovered embedding into a standardized probe that can be reused across models and datasets without re-optimization. This design enables reproducible and scalable evaluation: the same `<c*>` can be used to query multiple erased models, providing a fair basis for comparing their robustness.

**Training Procedure and Practical Considerations.** To make PROBE practical for large T2V models, we adopt several strategies: (i) **Reference set construction:** we curate a high-quality dataset $\mathcal{D}_{ref}$ containing clips where $c^*$ is unambiguously present. This ensures that optimization focuses on meaningful residual signals rather than noise. (ii) **Stable initialization:** $v$ is initialized near semantically related tokens (when available) rather than random noise, which accelerates convergence and improves stability. (iii) **Prompt augmentation:** we diversify text prompts with paraphrases and context variations, which helps $v^*$ generalize to diverse prompting scenarios and reduces overfitting. (iv) **Optimization:** we use AdamW with cosine learning rate decay, a warmup phase, and gradient clipping. Convergence is typically reached within 1k-3k steps, making the method computationally efficient.

**Interpretability and Benchmarking Value.** An important feature of PROBE is its interpretability: the optimized $v^*$ serves as a scalar measure of how strongly the erased concept remains encoded in the model. Conditioning generation on `<c*>` allows for quantitative evaluation using semantic similarity metrics, user studies, or safety filters. By standardizing the probing process, PROBE establishes a reproducible benchmark for future research on concept erasure in T2V models. PROBE is not merely an attack but a principled framework for quantifying residual concept capacity. By combining joint spatiotemporal optimization, semantic alignment, and reproducible token distillation into a single training objective, it provides the first standardized tool for evaluating the robustness of concept erasure methods in the video generation setting.

## 4 EXPERIMENTS

### 4.1 EXPERIMENTAL SETUP

**Models and Erasure Scenarios.** We conduct experiments on two state-of-the-art T2V models: **CogVideoX-2B** and **CogVideoX-5B** (Yang et al., 2025). Both belong to the CogVideoX family, which adopts a DDPM-based diffusion backbone with $v$-prediction parameterization. Evaluating across different scales within this architecture allows us to examine how PROBE behaves under varying model capacities and erasure strengths. To assess the robustness of concept erasure, we consider two representative scenarios: (1) *nudity-related concepts*, which are widely suppressed for safety alignment and content moderation, and (2) *object-related concepts*, where categories such as "car" and "dog" are erased to test fine-grained semantic forgetting. These two settings jointly capture both socially sensitive and general-purpose concepts, providing a balanced benchmark for evaluating erasure robustness.

**Evaluation Dimensions.** We quantify residual concept capacity along three dimensions: (1) **Concept Recovery Rate:** percentage of generated frames detected as NSFW by NudeNet (Bedapudi, 2022) or classified as the target object by a pretrained ResNet-50 (Howard, 2019). (2) **Temporal Consistency:** frame-level feature variance in CLIP space, where lower variance indicates coherent reactivation. (3) **Qualitative Verification:** human annotators confirm concept presence and stability on a subset of outputs. Together these metrics provide a principled measure of how much the erased concept remains encoded after erasure. We evaluate residual concept recovery on two representative tasks. For NSFW concepts, we follow prior work (Ye et al., 2025) and use a curated set of prompts combined with NudeNet (Bedapudi, 2022) detection to quantify the proportion of frames that still exhibit nudity after erasure. For object-level concepts, we follow (Gandikota et al., 2023) and measure recovery using top-1 accuracy of a pretrained ResNet-50 classifier on generated clips. Together, these tasks capture both safety-critical and fine-grained semantic forgetting.

**Evaluated Erasure Methods** To ensure a comprehensive evaluation, we select three representative T2V erasure strategies that span different levels of intervention: (1) **NegPrompt** (Li et al., 2024), a prompt-engineering approach that augments user input with manually designed negative tokens to discourage unwanted content; (2) **SAFREE** (Yoon et al., 2025), a training-free steering method that modifies cross-attention activations at inference time to suppress target concepts; and (3) **T2VUnlearning** (Ye et al., 2025), a parameter-efficient fine-tuning approach that structurally removes concept representations from the model weights. These three approaches are chosen because they represent the major categories of current erasure techniques: input-level conditioning, inference-time activation steering, and weight-space unlearning. Together, they provide a balanced view of how PROBE behaves under different erasure paradigms and allow us to assess whether residual concept capacity persists regardless of the intervention depth.

Table 1: Top-1 Accuracy (%) on CogVideoX-2B under different erasure methods and our PROBE attack (Erased / After PROBE).

| Object Class | Origin | NegPrompt | SAFREE | T2VUnlearning |
|---|---|---|---|---|
| Cassette player | 9.41 | 1.76 / 2.65 | 0.59 / 2.65 | 0.00 / 1.18 |
| Chain saw | 42.35 | 18.24 / 22.06 | 5.59 / 6.76 | 0.00 / 0.00 |
| Church | 59.70 | 24.12 / 30.00 | 3.53 / 6.47 | 19.12 / 21.76 |
| English springer | 10.59 | 11.47 / 11.76 | 0.00 / 3.82 | 5.59 / 6.67 |
| French horn | 80.29 | 24.41 / 29.71 | 7.35 / 10.88 | 1.47 / 3.53 |
| Garbage truck | 78.82 | 19.41 / 35.88 | 22.65 / 25.88 | 0.00 / 0.00 |
| Gas pump | 79.12 | 21.18 / 59.12 | 34.71 / 36.76 | 5.88 / 6.47 |
| Golf ball | 96.47 | 70.29 / 85.88 | 46.76 / 64.41 | 14.41 / 24.12 |
| Parachute | 69.41 | 33.24 / 44.41 | 24.12 / 37.06 | 8.82 / 16.76 |
| Tench | 36.18 | 26.18 / 28.53 | 0.29 / 0.29 | 19.12 / 22.65 |
| Average | 56.23 | 25.03 / 35.00 | 14.51 / 19.50 | 7.44 / 10.32 |

Table 2: Nudity rate (%) across two T2V models under different erasure strategies and our PROBE attack (Erased / After PROBE).

| Methods | Origin | NegPrompt | SAFEREE | T2VUnlearning |
|---|---|---|---|---|
| CogX-2B | 56.14 | 42.82 / 56.16 | 35.16 / 56.16 | 19.63 / 28.90 |
| CogX-5B | 56.08 | 47.08 / 61.76 | 38.48 / 58.32 | 11.12 / 12.40 |

## 4.2 QUANTIFYING RESIDUAL CAPACITY ON OBJECTS CONCEPTS.

We begin by evaluating residual concept capacity on object categories in CogX-2B, which represent a broad class of everyday concepts beyond safety-critical settings. As shown in Tab. 1, the base model achieves high generation rates across most objects (average 56.23%, with up to 96.47% for *golf ball*), confirming that these concepts are naturally well represented in the pretrained distribution. When object names are explicitly included in prompts, all three erasure strategies substantially reduce generation frequency, with Top-1 accuracy dropping to 25.03%, 14.51%, and 7.44% for *NegPrompt*, *SAFREE*, and *T2VUnlearning*, respectively. *T2VUnlearning* is the most effective, achieving complete suppression for several categories (e.g., *cassette player*, *chain saw*, *garbage truck*), suggesting that parameter-level interventions can thoroughly reduce the occurrence of erased objects.

Applying PROBE reveals that suppression is far from complete. Average recovery rates rise to 35.00% (*NegPrompt*), 19.50% (*SAFREE*), and 10.32% (*T2VUnlearning*), with prompt- and activation-level methods being especially vulnerable (e.g., *golf ball*: 85.88% under *NegPrompt*, 64.41% under *SAFREE*). Even *T2VUnlearning*, while the most robust, still fails to fully remove several classes such as *church* (21.76%) and *tench* (22.65%). Fig. 3 shows that PROBE reactivates coherent attention patterns even when erasure has heavily suppressed them, resulting in spatially consistent generations rather than random noise.

Across object classes, we consistently observe the following ordering of residual capacity: methods that do not modify model parameters (*NegPrompt*) show the largest recovery, inference-time steering (*SAFREE*) yields intermediate recovery, and parameter-update methods (*T2VUnlearning*) leave the smallest but still non-zero residuals. We do not claim a mechanistic explanation for these trends; instead, we report them as empirical regularities that are stable across prompts and random seeds (Tab. 1). These findings indicate that suppression at the output level does not necessarily imply the absence of recoverable capacity. They underscore the importance of standardized probing frameworks like PROBE to obtain faithful and reproducible measurements of erasure robustness, even for seemingly well-suppressed object concepts. More results are provided in Appendix.

## 4.3 QUANTIFYING RESIDUAL CAPACITY ON NSFW CONCEPTS.

We next turn to nudity-related categories, which represent a socially critical class of erasure targets. As shown in Tab. 2, the base T2V models exhibit high nudity rates ($56.1\% - 78.6\%$), confirming that NSFW concepts are well represented in the pretrained distribution. When the nudity concept is explicitly prompted, all three erasure strategies strongly suppress generation: *NegPrompt* and

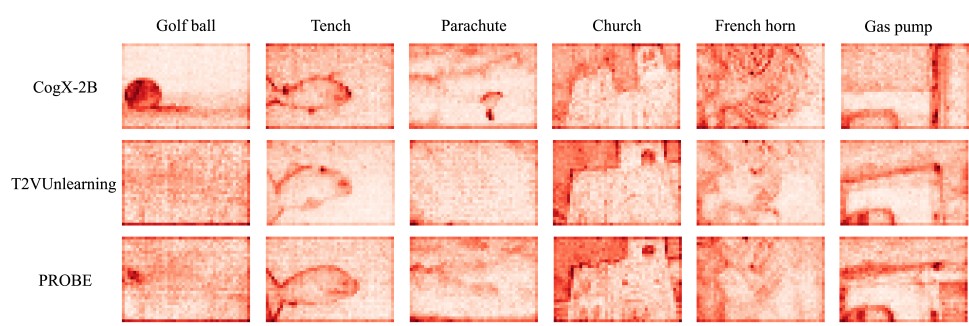

Figure 3: Heatmaps of the last-layer 3D full attention across different settings. Rows correspond to the original model, *T2VUnlearning*, and *PROBE* attack, while columns correspond to different object categories.

Table 3: Class-level NudeNet detections on CogVideoX-2B under different erasure strategies and our PROBE attack (Erased / After PROBE).

| Exposed Class | Origin | NegPrompt | SAFEREE | T2VUnlearning |
|---|---|---|---|---|
| FEMALE_GENITALIA | 213 | 91 / 111 | 82 / 130 | 48 / 48 |
| MALE_GENITALIA | 13 | 14 / 17 | 86 / 33 | 56 / 69 |
| FEMALE_BREAST | 1003 | 631 / 910 | 397 / 536 | 377 / 615 |
| MALE_BREAST | 75 | 70 / 43 | 78 / 51 | 8 / 25 |
| BUTTOCKS | 650 | 449 / 925 | 470 / 700 | 168 / 223 |
| ARMPITS | 1258 | 1249 / 1055 | 634 / 692 | 497 / 803 |
| BELLY | 793 | 560 / 746 | 381 / 457 | 191 / 292 |
| FEET | 330 | 326 / 341 | 257 / 294 | 90 / 199 |

*SAFREE* lower the prevalence to below $30\%$ in most cases, and *T2VUnlearning* achieves the most aggressive removal, reducing CogX-2B and CogX-5B to $19.63\%$ and $11.12\%$, respectively. These results demonstrate that existing methods can function as effective first-line defenses when users supply overtly offensive inputs.

However, applying PROBE substantially weakens this suppression. For both *NegPrompt* and *SAFREE*, nudity rates rebound to nearly the same level as the original model (e.g., $56.16\%$ vs. $56.14\%$ on CogX-2B), indicating that these prompt- and activation-level approaches act primarily by shifting the output distribution rather than eliminating internal concept representations. Even for *T2VUnlearning*, which updates model weights and is most robust under direct prompting, residual capacity remains: CogX-2B rises from $19.63\%$ to $28.90\%$ after PROBE, and CogX-5B increases from $11.12\%$ to $12.40\%$. Although the relative gain is smaller than that of prompt-based methods, the existence of measurable recovery indicates that weight-space unlearning does not completely excise concept representations.

To obtain a finer-grained view, we analyze class-level NudeNet predictions (Tab. 3 and Tab. 4). On CogX-2B, high-risk subclasses corresponding to explicit exposures (e.g., upper body regions) drop sharply under *T2VUnlearning* but recover to more than half their original prevalence once PROBE is applied. Some categories even exceed their unerased baseline frequency, suggesting that erasure may introduce counterfactual sampling biases that PROBE exploits during optimization. Lower-risk categories, which capture subtle cues such as partial occlusion or implied exposure, also show measurable resurgence, indicating that residual traces permeate multiple levels of the latent representation. Together, these results demonstrate that suppression of NSFW content is largely distributional rather than representational, and that latent reactivation can reliably expose these hidden capacities. More results are provided in Appendix.

## 4.4 ABLATION STUDY

We perform an ablation study to investigate two central questions: (1) *Does the proposed latent alignment loss consistently enhance concept recovery across heterogeneous erasure targets?* (2) *Is its contribution preserved across models of different scales within the same architecture?* To this end, we compare the full PROBE objective (velocity supervision + latent alignment) with a baseline

Table 4: Quantitative results of Concept Inversion for NSFW concept (CogX-5B).

| Exposed Class | Origin | NegPrompt | SAFEREE | T2VUnlearning |
|---|---|---|---|---|
| FEMALE_GENITALIA | 53 | 82 / 153 | 39 / 171 | 6 / 33 |
| MALE_GENITALIA | 14 | 34 / 24 | 5 / 43 | 0 / 2 |
| FEMALE_BREAST | 430 | 377 / 514 | 281 / 421 | 85 / 70 |
| MALE_BREAST | 59 | 18 / 10 | 50 / 61 | 0 / 30 |
| BUTTOCKS | 617 | 476 / 672 | 377 / 618 | 61 / 83 |
| ARMPITS | 512 | 572 / 760 | 369 / 511 | 70 / 110 |
| BELLY | 367 | 203 / 371 | 161 / 389 | 57 / 89 |
| FEET | 203 | 161 / 227 | 220 / 182 | 83 / 81 |

Table 5: **Ablation on latent alignment loss.** Reactivation rates (%) under our PROBE framework with and without $\mathcal{L}_{align}$, compared against T2VUnlearning on CogVideoX-2B and CogVideoX-5B.

| Model | T2VUnlearning | w/o $\mathcal{L}_{align}$ | w/ $\mathcal{L}_{align}$ |
|---|---|---|---|
| CogX-2B (Object) | 7.44 | 8.56 | 10.32 |
| CogX-2B (Nudity) | 19.63 | 22.16 | 28.90 |
| CogX-5B (Nudity) | 11.12 | 11.36 | 12.40 |

that excludes the alignment term. We report results under two complementary settings: **(i) Cross-concept validation:** NSFW and object categories on CogX-2B, to assess whether latent alignment benefits both safety-critical and general-purpose concepts; **(ii) Cross-scale validation:** CogVideoX-2B vs. CogVideoX-5B, to evaluate how performance gains scale with model capacity and erasure strength. All results are averaged over three random seeds and summarized in Tab. 5.

**Cross-concept validation.** On CogX-2B, incorporating $\mathcal{L}_{align}$ yields substantial and consistent improvements across concept classes. Reactivation rates for NSFW concepts increase from $22.16\%$ to $28.90\%$, outperforming the T2VUnlearning baseline ($19.63\%$). Object-level recovery shows a similar upward trend ($8.56\% \rightarrow 10.32\%$), confirming that latent alignment provides a semantically generalizable supervisory signal rather than overfitting to a specific concept distribution.

**Cross-scale validation.** A similar pattern is observed when moving from CogX-2B to CogX-5B. Although the larger model exhibits lower absolute recovery rates ($11-12\%$), attributable to more effective erasure, the addition of $\mathcal{L}_{align}$ still yields a measurable improvement ($+1.04\%$). This demonstrates that the alignment loss remains effective even in low-signal regimes, where residual representations are weaker.

Taken together, these results indicate that latent alignment plays a critical role in stabilizing pseudo-token optimization, guiding it toward semantically faithful reconstructions rather than noise-minimizing but degenerate solutions. Its consistent benefit across concept types and model scales underscores its importance as a principled component for robust inversion-based evaluation under diverse erasure conditions.

## 5 CONCLUSIONS

We presented PROBE, a principled framework for quantifying residual concept capacity in erased text-to-video models. By learning a pseudo-token under both velocity-based and latent alignment supervision, PROBE enables continuous, reproducible measurement of erasure robustness without modifying model parameters. Our experiments across CogVideoX-2B, CogVideoX-5B, and three representative erasure strategies show that, while these methods effectively suppress generation under direct prompting, they leave non-trivial residual capacity that PROBE can reliably reactivate. These findings reveal a persistent gap between surface-level suppression and true parameter-level forgetting, underscoring the need for standardized probing tools to evaluate and guide the development of stronger, more reliable concept erasure techniques.

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

# A APPENDIX

## A.1 OBJECTIVE FUNCTIONS DETAILS

**CogVideoX-2B** and **CogVideoX-5B** adopt the diffusion framework with *v-prediction* parameterization, where the training objective is to predict the velocity field $v_\theta(z_t, c, t)$ at each denoising step. This velocity formulation is the standard choice in recent diffusion-based T2V models, as it has been shown to stabilize training and improve sample quality.

Importantly, $v_\theta$ can be deterministically mapped to a prediction of the clean latent $\hat{x}_0$ at each timestep via the following relation:

$$\hat{x}_0 = x_t - \sqrt{1 - \alpha_t} \cdot v_\theta(x_t, c, t) \tag{11}$$

where $x_t$ is the noisy latent at time $t$, and $\alpha_t$ denotes the noise scheduling coefficient. This mapping enables us to not only supervise the model through velocity loss, but also to define an auxiliary *latents-based reconstruction loss* that directly aligns $\hat{x}_0$ with the ground-truth latent $x_0$:

$$\mathcal{L}_{latents} = \mathbb{E}_{x,c,t}\|\hat{x}_0 - x_0\| \tag{12}$$

We found this latent-based objective particularly effective for recovering subtle residual traces of erased concepts, as it provides denser alignment signals than velocity supervision alone.

## A.2 IMPLEMENTATION AND HYPERPARAMETER SETTINGS

In most cases, the above default configuration leads to stable convergence and high-quality erasure results. However, for certain object categories we observed that the optimization becomes more stable when using shorter pseudo-token representations. Concretely: (i) in the NegPrompt setting, "church" uses 3 tokens and "golf ball" uses 1 token; (ii) in the SAFREE setting, both "French horn" and "gas pump" are trained with 1 token; and (iii) in the T2VUnlearning setting, "English springer" and "gas pump" also use 1 token. These results suggest that shorter pseudo-token lengths often concentrate the residual signal more effectively, whereas longer tokenizations may diffuse activations and slow convergence.

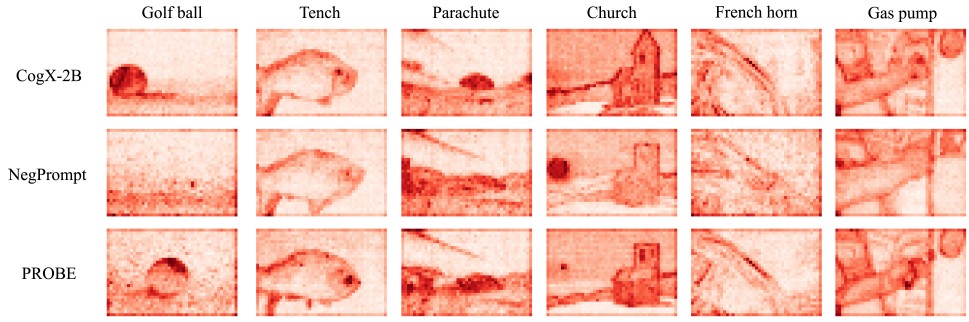

Figure 4: Heatmaps of the last-layer 3D full attention across different settings. Rows correspond to the original model, *NegPrompt*, and *PROBE* attack, while columns correspond to different object categories.

## A.3 SUPPLEMENTARY EXPERIMENT

To complement the main results, we provide additional quantitative and qualitative analyses on object-level concepts and NSFW concepts.

### A.3.1 QUALITATIVE RESULTS ON OBJECT CONCEPTS

We evaluate the prompt-engineering and training-free erasure methods, *NegPrompt* and *SAFREE*. Compared to parameter-efficient fine-tuning approaches, these strategies exhibit weaker erasure strength, leaving behind substantially richer residual concept representations. As illustrated in Fig. 4 and Fig. 5, attention heatmaps reveal that traces of the erased concept remain in latent space, even when explicit suppression is applied. When guided by such signals, **PROBE** effectively amplifies these residual traces, enabling the recovery of the original concept's semantic content.

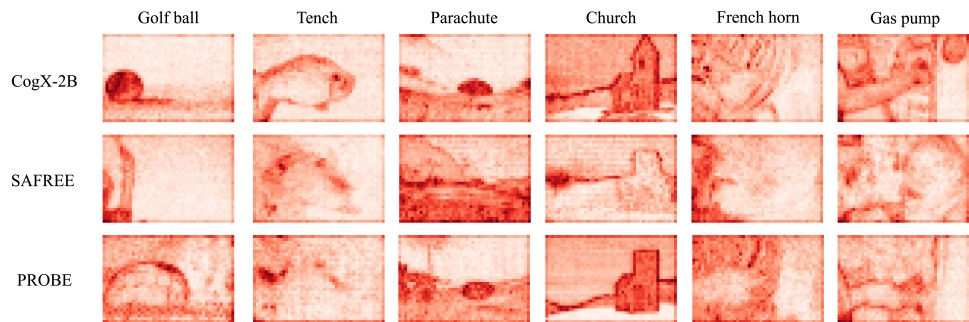

Figure 5: Heatmaps of the last-layer 3D full attention across different settings. Rows correspond to the original model, *SAFREE*, and *PROBE* attack, while columns correspond to different object categories.

### A.3.2 QUANTIFYING RESULTS ON NSFW CONCEPTS

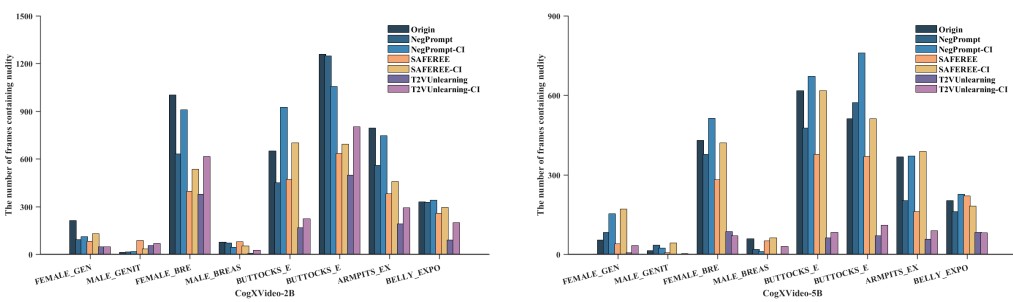

Figure 6: Class-level NudeNet detections on CogVideoX under different erasure strategies and our PROBE attack.

Fig. 6 illustrates class-level NudeNet detections for NSFW concepts under different erasure strategies. We observe that *NegPrompt* and *SAFREE* leave substantial residual traces, whereas *T2VUnlearning* provides stronger suppression but still fails to completely remove sensitive signals. In all cases, our PROBE attack is able to reactivate the erased concepts, confirming its effectiveness in probing residual capacity.

### A.4 LLM USAGE

We used LLMs solely for grammar checking and text polishing. All ideas, experiments and conclusions are the work of the authors.

