# OpenReview forum: "PROBE: Probing Residual Concept Capacity in Erased Text-to-Video Models"
_ICLR.cc/2026/Conference — ICLR 2026 Conference Withdrawn Submission_

### Official Review · Reviewer_eMjV · 2025-10-25

**Soundness:** 1
**Presentation:** 2
**Contribution:** 1
**Rating:** 2
**Confidence:** 4

**Summary:**

This paper proposes a learning-based framework designed to induce erased text-to-video (T2V) diffusion models to regenerate previously unlearned target concepts. Building upon Textual Inversion, the method learns a special token representing the target concept with the proposed latent alignment loss using a reference dataset.
Experiments on two T2V models (CogVideoX-2B and CogVideoX-5B) combined with three different T2V concept-erasure methods show that the proposed method can partially restore erased concepts.

**Strengths:**

- The proposed method is simple.
- The paper is mostly clear and well-written.

**Weaknesses:**

- It is unclear why existing adversarial attack or prompt manipulation methods developed for text-to-image (T2I) models, such as P4D [1], Ring-A-Bell, MMA [3], UnlearnDiff [4], cannot be directly applied to T2V models. These methods also aim to elicit erased or undesired content, and a discussion or comparison with them would strengthen the motivation.

- The process of curating the reference set $\mathcal{D}_{ref}$ is unclear. Since it is used to learn the special token, it’s important to know the details, e.g., the total duration and the required quality or diversity of the video clips.
- Both evaluated models belong to the CogVideoX family, limiting the claim that the learned token $v*$ can be reused for any T2V models. Including results from other T2V models, e.g., LTX-Video [5] and Open-Sora [6], would strengthen the evaluation and demonstrate cross-model transferability of the learned token $v*$.
- Although the paper said that the temporal consistency (frame-level CLIP feature variance) and qualitative human evaluation are also included in evaluation dimensions, these metrics are not reported in the tables. Therefore, it’s unclear whether the proposed method affects visual quality, generation diversity, or temporal coherence of the erased T2V model.
- The method’s effectiveness appears limited. When applied to the strongest erasure method (T2VUnlearning), it improves accuracy by less than 3% in Table 1 and only 1.28% for CogVideoX-5B in Table 2. Also, results for object-related concepts are reported only for CogVideoX-2B, with no corresponding experiments presented for CogVideoX-5B.

- (minor) More training details should be provided, including the learning rate, gradient clipping threshold, and the value of the loss-balancing coefficient $\lambda$.
- (minor) The main figure (Fig. 2) is not well designed. It doesn’t contain sufficient information about the proposed method and occupies a large space. The diffusion model, the reference set, the ground-truth latents, and the predicted latents can be shrunk since they are not the main focus of this paper. Moreover, adding notations in the figure could provide better clarity, e.g., $z_0^{(i)}$, $z_t^{(i)}$, and $\hat{z}_0^{(i)}(v, t)$.

[1] Prompting4Debugging: Red-Teaming Text-to-Image Diffusion Models by Finding Problematic Prompts. Zhi-Yi Chin et al. [ICML 2024]
[2] Ring-A-Bell! How Reliable are Concept Removal Methods for Diffusion Models? Yu-Lin Tsai et al. [ICLR 2024]
[3] MMA-Diffusion: MultiModal Attack on Diffusion Models. Yijun Yang et al. [CVPR 2024]
[4] To Generate or Not? Safety-Driven Unlearned Diffusion Models Are Still Easy To Generate Unsafe Images ... For Now. Yimeng Zhang et al. [ECCV 2024]
[5] LTX-Video: Realtime Video Latent Diffusion. Yoav HaCohen et al. [arXiv:2501]
[6] Open-Sora. https://github.com/hpcaitech/Open-Sora.

**Questions:**

Please see the weaknesses above.

---

### Official Review · Reviewer_9i28 · 2025-10-27

**Soundness:** 2
**Presentation:** 2
**Contribution:** 2
**Rating:** 4
**Confidence:** 3

**Summary:**

The paper proposes PROBE, a framework to quantify how much of a target concept remains in a text-to-video diffusion model after an erasure or safety-guarding procedure. PROBE learns a compact pseudo-token by optimizing a shared embedding across frames and timesteps and adds a latent-alignment objective to encourage clip-level semantic faithfulness. The learned token becomes a reusable probe for standardized robustness evaluation across models and erasure methods. Experiments on CogVideoX-2B/5B and three representative erasure strategies show that measurable residual capacity persists post-erasure.

**Strengths:**

- Timely problem formulation for video, where temporal dependencies can reintroduce erased concepts; the paper articulates these video-specific challenges well.
- Clear, modular method with an interpretable probe token that standardizes evaluation across models and erasure methods.

**Weaknesses:**

1. While the paper covers safety-sensitive and object categories, the generality to other safety domains (e.g., violence, celebrities, logos/copyrighted entities) is not demonstrated here.
2. Cost and scalability: although step counts are given, detailed wall-clock and resource scaling with clip length, resolution, and model size would help assess deployability.
3. The paper shows heatmaps and aggregate rates, but doesn’t dig into where PROBE fails (which concepts or motion patterns resist reactivation), or when it over-reactivates unrelated semantics. Understanding limits could guide defense design.
4. Results are mostly under in-distribution prompt families and specific models. How does PROBE transfer across datasets, durations (longer clips), camera motions, or negative contexts (e.g., “a beach with no people”)? The token’s portability is a key claim; more cross-domain tests would help.
5. Robustness to adversarial or diverse paraphrases remains uncertain. Reporting performance under adversarially optimized prompts (beyond long/refined ones) would be useful.
6. Related works of concept erasure in T2I models are outdated. Incorporating more recent works [1,2,3] would be more complete.

[1] EraseAnything: Enabling Concept Erasure in Rectified Flow Transformers

[2] Erased or Dormant? Rethinking Concept Erasure Through Reversibility

[3] Set You Straight: Auto-Steering Denoising Trajectories to Sidestep Unwanted Concepts

**Questions:**

See weaknesses

---

### Official Review · Reviewer_xrN5 · 2025-10-31

**Soundness:** 2
**Presentation:** 3
**Contribution:** 1
**Rating:** 2
**Confidence:** 3

**Summary:**

This paper aims to regenerate erased concepts in text-to-video models. The authors propose training a shared token embedding across frames by aligning latent representations with the embeddings of the target concept. Experiments on the erasure of nudity and object-related concepts indicate that the proposed method can act as an attack to regenerate erased concepts against concept-erasure techniques.

**Strengths:**

1. The latent alignment used in this approach leverages shared latent embeddings across different models, which potentially enables transferability among them.
2. The proposed method considers the characteristics of videos and integrates them into the optimization objective.

**Weaknesses:**

1. The claimed contributions of this paper appear to be somewhat overstated. The authors argue that PROBE constitutes a principled framework for quantifying residual concept capacity. However, in practice, PROBE primarily functions as an attack mechanism that reconstructs erased concepts rather than a truly principled measurement framework. First, although the latent alignment loss aims to enhance semantic consistency, it cannot ensure that the recovered signal reflects true residual concept capacity, as the optimization mainly amplifies superficial cues from reconstruction and latent correlations. Second, since model parameters, attention maps, and other internal mechanisms remain unaltered, PROBE effectively behaves as a token-level adversarial probe rather than a comprehensive diagnostic tool.

2. The experimental setup is unclear. In Table 1 and 2, is PROBE optimized on one erasure method and then transferred to attack others, or is it separately optimized for each erasure method? And in Table 5, is PROBE optimized on CogX-2B (Object) and then transferred to attack others? This paper also claimed that the resulting embedding is a reusable probe across different erasure methods and models, but it is unclear which experiments or tables demonstrate this reusable performance.

3. PROBE’s performance heavily depends on the data selection for the regenerated concept, yet the authors do not discuss this influence, limiting its applicability.

**Questions:**

1. Clarification on claimed contributions
2. Experimental setup and transferability
3. Influence of data selection

---

### Official Review · Reviewer_jr5B · 2025-11-06

**Soundness:** 2
**Presentation:** 2
**Contribution:** 2
**Rating:** 4
**Confidence:** 4

**Summary:**

This paper presents PROBE, a method to quantify residual concept capacity in text, to, video (T2V) diffusion models after concept erasure.
The idea is to optimize a pseudo, token embedding that reactivates an erased concept when inserted into prompts. Unlike standard textual inversion, PROBE performs joint spatiotemporal optimization and introduces a latent alignment loss to maintain temporal coherence. The learned embedding is later distilled into a reusable “probe token” to benchmark erasure methods.

Experiments on CogVideoX, 2B and 5B models with NegPrompt, SAFREE, and T2VUnlearning erasure strategies show that erased concepts—both object, level and NSFW—can still be partially recovered. The authors argue that current unlearning techniques suppress rather than remove representations and propose PROBE as a standardized evaluation tool for erasure completeness.

**Strengths:**

Relevance and timeliness: Addresses an urgent open issue — whether “concept erasure” in T2V models truly eliminates internal representations or only suppresses them.

Methodological soundness: The proposed spatiotemporal pseudo, token optimization and latent, alignment loss are sensible and practically valuable extensions of textual inversion to video.

Empirical breadth: The study covers multiple erasure techniques and model scales, consistently demonstrating measurable recovery of erased concepts.

Diagnostic utility: The concept of a reusable “probe token” provides a simple, reproducible framework for future evaluation of model erasure and unlearning methods.

Writing and figures: The paper is clearly structured, visually polished, and easy to follow.

**Weaknesses:**

1. Limited novelty.
The core idea largely builds on known textual inversion techniques with minor adaptations (temporal optimization and latent alignment). These are incremental engineering extensions, not conceptual innovations. The paper lacks theoretical justification for why these adaptations uniquely capture residual capacity in erased models.

2. Weak methodological rigor and incomplete reporting.
Key hyperparameters (learning rate, λ grid, token length, reference set size, seeds, clip length) are not specified, making the work difficult to reproduce. Statistical measures such as variance, confidence intervals, or hypothesis testing are absent, weakening the empirical support for the main claims.

3. Evaluation metrics and human study insufficiently justified.
The use of ResNet, 50 and NudeNet as recovery metrics is limited and prone to false positives/negatives. These detectors are not temporally aware, and the paper does not provide an analysis of their reliability. Human evaluation is briefly mentioned but lacks methodological transparency (number of annotators, criteria, agreement scores).

4. Missing mechanistic insight.
The study confirms that residual capacity exists but does not explore where it resides (e.g., cross, attention vs. U, Net layers). The qualitative visualizations shown are descriptive but not explanatory. A deeper interpretability analysis would strengthen the work’s scientific impact.

5. Missing baselines and ablations.
The paper targets three erasure methods, but it omits several relevant comparisons: e.g., text, encoder editing methods (some are cited in related work), attention, pruning approaches, or stronger adversarial/paraphrase attacks adapted from T2I. It is unclear whether PROBE is strictly stronger than naïve textual inversion per, frame or per, timestep. Ablations on reference set size, token length choices (A.2 hints at token lengths but lacks systematic study), and prompt augmentation choices are needed to quantify which design choices matter most.

6. Ethical and dual, use concerns.
The method demonstrates recovery of NSFW content from erased models but lacks a clear discussion of ethical implications, responsible disclosure, or mitigation strategies. This omission is significant given the safety implications of reactivating restricted content.

**Questions:**

Questions for Authors

1) How sensitive is PROBE’s success to the λ hyperparameter, reference set size, and token length? Please include quantitative ablations if possible.

2) Does PROBE recover semantically similar alternatives (e.g., “nude” → “bare”) or the exact erased concept? How is this verified?
How does PROBE compare to per, frame textual inversion or simple CLIP, guided optimization?

3) Can the learned pseudo, token transfer between model scales (e.g., from CogVideoX, 2B → 5B)?

4) Could the authors clarify whether the probe embeddings or code will be released, and if so, under what ethical restrictions?

5) Please elaborate on the human evaluation setup (number of evaluators, sampling process, agreement statistics).

---

### Note · Authors · 2025-11-14

I have read and agree with the venue's withdrawal policy on behalf of myself and my co-authors.